# Dual Input Fuzzy Logic Controllers for Closed Loop Hemorrhagic Shock Resuscitation

**David Berard** [1], **Saul J. Vega** [1], **Guy Avital** [1,2,3] and **Eric J. Snider** [1,*]

1 U.S. Army Institute of Surgical Research, JBSA Fort Sam Houston, San Antonio, TX 78234, USA
2 Trauma & Combat Medicine Branch, Surgeon General's Headquarters, Israel Defense Forces, Ramat-Gan 52620, Israel
3 Division of Anesthesia, Intensive Care & Pain Management, Tel-Aviv Sourasky Medical Center, Sackler Faculty of Medicine, Tel-Aviv University, Tel-Aviv 64239, Israel
* Correspondence: eric.j.snider3.civ@health.mil; Tel.: +1-210-539-8721

**Abstract:** Hemorrhage remains a leading cause of preventable death in emergency situations, including combat casualty care. This is partially due to the high cognitive burden that constantly adjusting fluid resuscitation rates can require, especially in austere or mass casualty situations. Closed-loop control systems have the potential to simplify hemorrhagic shock resuscitation if properly tuned for the application. We have previously compared 4 different controller types using a hardware-in-loop test platform that simulates hemorrhagic shock conditions, and we found that a dual input—(1) error from target and (2) rate of error change—fuzzy logic (DFL) controller performed best. Here, we highlight a range of DFL designs to showcase the tunability the controller can have for different hemorrhage scenarios. Five different controller setups were configured with different membership function logic to create more and less aggressive controller designs. Overall, the results for the different controller designs ranged from reaching the setup rapidly but often overshooting the target to more conservatively approaching the target, resulting in not reaching the target during high active hemorrhage rates. In conclusion, DFL controllers are well-suited for hemorrhagic shock resuscitation and can be tuned to meet the response rates set by clinical practice guidelines for this application.

**Keywords:** control systems; hemorrhagic shock; fluid resuscitation; fuzzy logic; closed-loop; fluid resuscitation; hardware-in-loop





## 1. Introduction

Hemorrhage is the most common cause of preventable death in both civilian [1] and military [2] trauma casualties. The main pillars of care for these patients are expeditious hemorrhage control and volume resuscitation—the restoration of blood volume, preferentially using whole blood or blood components, to restore oxygen delivery to the end organs [3]. In cases where definite control of the hemorrhage is not immediately achievable, most experts recommend the "damage control resuscitation" (DCR) approach, which prompts goal-directed volume resuscitation balancing the need for restoring perfusion on one hand, while avoiding exacerbation of the hemorrhage on the other [3]. However, this can require constant monitoring of the patient's condition and frequent adjusting of the infusion rate.

As this task can be described as controlling a variable (e.g., blood pressure) towards a setpoint (i.e., the resuscitation goal), it is not surprising that several attempts have been made to automize this task in a closed-loop controlled fashion [4]. They vary in the approaches taken, secondary to the intended use case. A variety of approaches, including complex mathematical modeling [5,6] and adaptive controls [7,8] were described for the purpose of hemodynamic control through fluid management. However, DCR in its most basic form, which resembles current clinical (manual) practice, can be described as a single input (e.g., blood pressure)—single output (infusion flow rate). Hence, simpler

controllers, such as decision tables, proportional-integral-derivative (PID) and fuzzy logic (FL) controllers [9] should at least be considered for this purpose.

We have previously developed a hardware-in-loop automated test platform for resuscitation controllers (HATRC) for comparing the performance between closed-loop controller designs across a wide range of hemorrhage resuscitation scenarios [10]. With this, we recently compared various controller logic types and determined that a dual-input fuzzy logic controller design performed best [11]. This was determined across various subject variability runs and four hemorrhage scenarios, using aggregate performance metrics tied to the intensity of the resuscitation, stability of the subject, and resource efficiency. In this work, we expand on this previous study to compare a range of dual-input fuzzy logic controller types to highlight the controller capabilities based on tuning for hemorrhagic shock resuscitation.

## 2. Materials and Methods

### 2.1. Overview of HATRC Platform

We previously developed the Hardware-in-loop Automated Testbed for Resuscitation Controllers (HATRC) for the purpose of high throughput testing of physiological closed-loop controllers designed to control fluid infusion, particularly for hemorrhagic shock resuscitation [10,12]. Water was circulated in a closed-loop by a peristaltic pump (Masterflex L/S, Masterflex Bioprocessing, Vernon Hills, IL, USA) while pressure was monitored and recorded using LabChart (PowerLab, ADinstruments, Sydney, Australia) via pressure transducer (ICU Medical, San Clemente, CA, USA). A key component of the system was the PhysioVessel (PV) model, a customizable fluidic reservoir that provides a volume-responsive hydrostatic pressure [13]. Analysis of a large animal hemorrhage model revealed a linear pressure–volume response for the administration of whole blood in swine who underwent a spleen injury following a controlled hemorrhage. Though alternative pressure–volume curves were found to characterize other fluids, like crystalloids, only whole blood was used as the simulated infusate during the hemorrhage scenarios in this study. The whole blood-tuned PV ($PV_{WB}$) was connected to two additional peristaltic pumps. One pump provided outflow comprised of a basal urine rate and a hemorrhage rate determined by the current hemorrhage scenario (see Section 2.2). The other pump provided an infusion whose rate was controlled by the resuscitation controller being evaluated. MATLAB (MathWorks, Natick, MA, USA) was used to run the hemorrhage scenario, determine inflow rates based on resuscitation controller algorithms, and control the corresponding pumps through an RS232 USB-to-serial adapter (CoolGear, Clearwater, FL, USA) configured as indicated by the pumps' manufacturer.

### 2.2. Hemorrhage Scenarios for Controller Performance

For a previous study, we designed 11 simulated hemorrhage scenarios to evaluate the performance of fluid resuscitation controllers by challenging them to operate against a variety of bleeding rates and initial arterial pressures [12]. Given the similarities found in controllers' performances in several scenarios during that study, here we focus on four distinct whole-blood scenarios to assess the new set of fuzzy logic controllers. Throughout, a target pressure of 65 mmHg mean arterial pressure (MAP) was the goal controllers were seeking during resuscitation.

Scenario 1 was the only scenario to last 62 min, and it simulated a compressible bleed that was already under control by the time resuscitation started. During the first half of this scenario, the fluid controller attempted to resuscitate the simulated subject from an initial MAP of 45 mmHg up to a target MAP of 65 mmHg without an active hemorrhage. At the 30-min mark, however, a high-rate bleed lasting 2 min was triggered, simulating a loosening and re-tightening of a tourniquet. Afterwards, hemorrhage was stopped, and controllers were given an additional 30 min to restabilize at target MAP.

The remaining three scenarios all lasted 30 min and simulated non-compressible hemorrhages. Both Scenarios 2 and 3 allowed natural coagulation to affect the simulation—the

only difference was in the initial MAP: Scenario 2 started in a state of simulated compensated shock at 65 mmHg, while Scenario 3 started in a state of decompensation at 45 mmHg. Finally, Scenario 4 mimicked a subject with an initial MAP of 45 mmHg, who starts to experience a gradual degradation of their internal hemostatic mechanisms 5 min into the resuscitation.

### 2.3. Fuzzy-Logic Controller Design

Fuzzy logic controllers are widely used in industries such as manufacturing [14,15], automobile operation [16–18], and even space exploration [19,20], and their utility in various areas of medical care has been a subject of ongoing research and development [21–24]. Fuzzy logic takes a discrete input value and classifies it into a non-discrete linguistic, or descriptive, term using a set of membership functions. These functions map the input to a value of 0–1 which is its degree of membership for each class within the linguistic set. A set of logical rules then evaluate the fuzzified input(s) to determine the corresponding output. This approach is particularly advantageous when precise classifications are not easily determined, making Boolean-based logic suboptimal. The nonlinear, time-varying nature of the cardiovascular system makes it a prime candidate for fuzzy logic control.

We previously tested multiple types of hemorrhagic shock resuscitation controllers on HATRC that included two different versions of decision table, PID, single-input fuzzy logic (SFL), and dual-input fuzzy logic (DFL) controllers. Based on a comparative analysis using select controller performance metrics and a set of three aggregate metrics, described in further detail in Section 2.4, we determined that the DFL controllers demonstrated the best balance of Intensity, Stability, and Resource Efficiency [12]. We kept the two original DFL controller configurations and included an additional three DFL controllers with a wider range of tuning variations for a total of 5 in a comparative study using HATRC. The MATLAB Fuzzy Logic Designer toolbox was used to develop all the controllers evaluated here, and the infusion flow rate was the single output to the system. The first input to the controllers was the error expressed as a percentage of the measured system pressure divided by the setpoint, with a value of 1 representing the target being reached and was titled *PerformanceError* (Figure 1). The second input was the rate of change in the error over time taken as the slope of a linear regression across the last three samples and was titled *(d/dt)PerformanceError* (Figure 1).

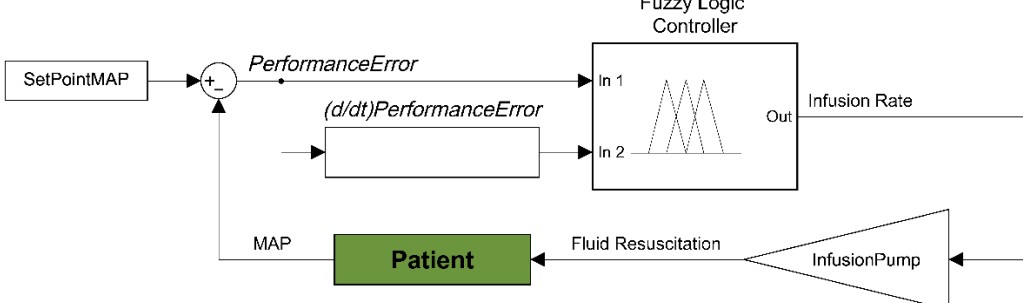

**Figure 1.** Diagram of the dual input fuzzy logic controller for hemorrhagic shock resuscitation. Two inputs to the fuzzy logic controller are derived from input pressure readings and the distance from set point mean arterial pressure—performance error and rate of performance error change. These two inputs are used to determine an infusion rate output for providing fluid to resuscitate and stabilize pressure. Each controller was set with the same types of membership functions, but with varying constants. DFL 1−4 classified *PerformanceError* into three fuzzy sets: *VeryLow*, *Low*, and *Set*. DFL 5 used these same three with an additional set called *Over*. All controllers used z-shaped membership functions for mapping *PerformanceError* into *VeryLow* and s-shaped membership functions for *Set*. DFL 1-3 used simple Gaussian curves while 4 and 5 used generalized bell-shaped membership functions for mapping *PerformanceError* into *Low*. DFL 5 also used an s-shaped membership function

to map *Over*. Smooth and Gaussian curves were selected as the membership functions for input 1 due to their lower computational cost and guaranteed continuity when compared to trapezoidal functions [25,26]. They have also been shown to be easier to optimize using evolutionary computational algorithms in type-2 fuzzy controllers which will be important for future iterations. Parameters for the functions used here were informed by expert feedback and current DCR guidelines. The membership functions for both inputs of all controllers are shown in Figure 2.

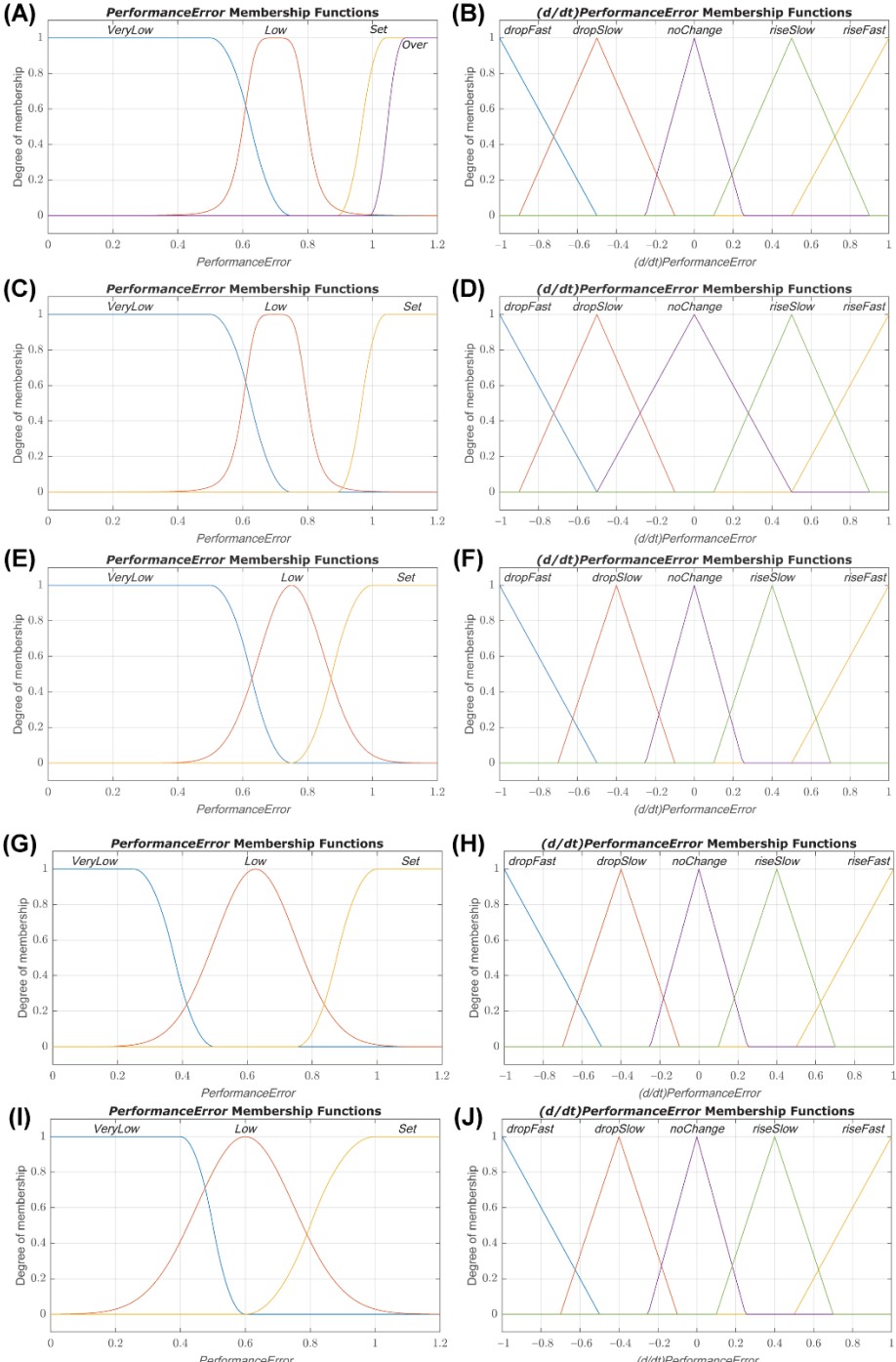

**Figure 2.** Membership function plots for dual-input fuzzy logic controllers. Plots of the *PerformanceError* and *(d/dt)PerformanceError* input membership functions for dual-input fuzzy logic controllers 1 (**A,B**), 2 (**C,D**), 3 (**E,F**), 4 (**G,H**), and 5 (**I,J**).

All controllers used the same fuzzy sets and membership function types for *(d/dt)PerformanceError*. Five fuzzy sets were defined: *dropFast*, *dropSlow*, *noChange*, *riseSlow*, and *riseFast*. Linear z-shaped membership functions were used to map *(d/dt)PerformanceError* into *dropFast*, triangular membership functions were used for *dropSlow*, *noChange*, and *riseSlow*, and linear s-shaped membership functions were used to map *riseFast*. Parameters of the membership functions for both inputs were tuned for each controller to produce a range of performance (e.g., prioritizing reaching the set point quickly vs. prioritizing minimum overshoot of the set point). Distinct rules were created for each controller using a similar ethic, and plots of the resulting rule surfaces are presented in Figure 3. The output, titled *InfusionRate*, was broken into the fuzzy sets *Off*, *Med*, and *Max* which utilized linear functions mapping to the output values of 0, 250 mL/min, and 500 mL/min, respectively. All controllers were type-1 Sugeno systems and used the following implication methods: a product AND, probabilistic OR, minimum Implication, maximum Aggregation, and a weighted average defuzzification method.

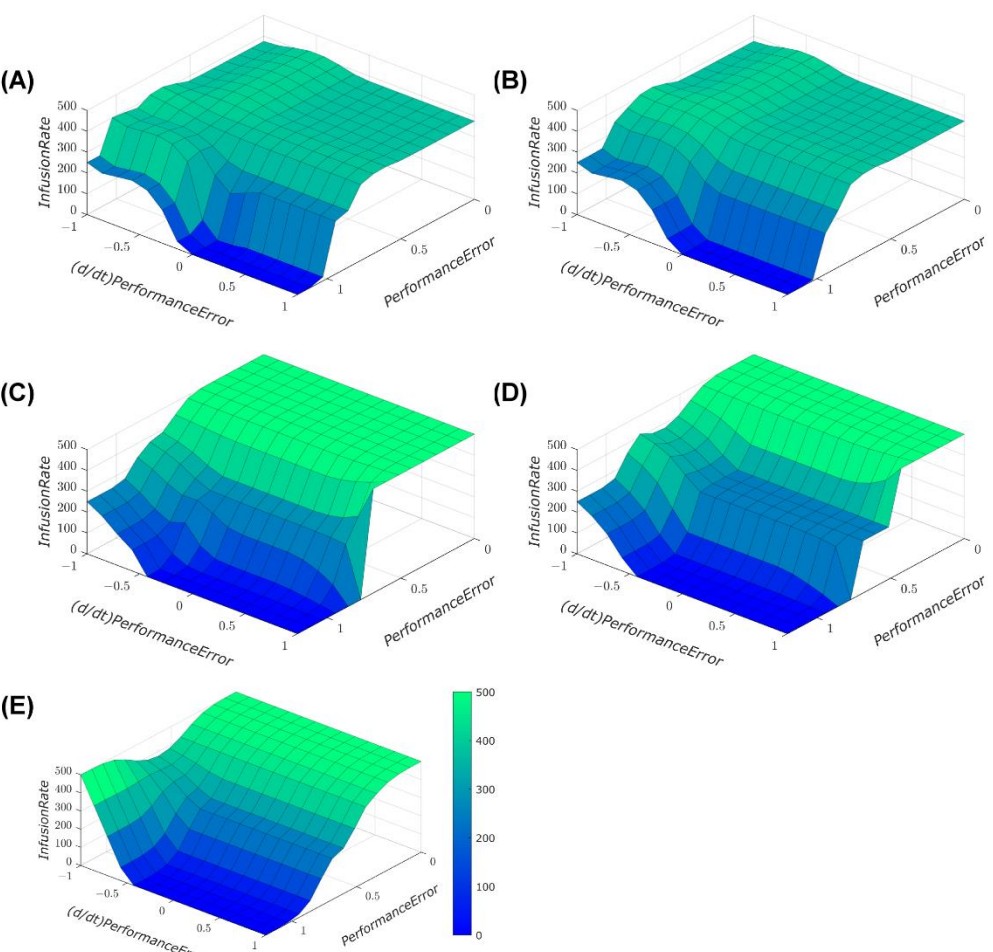

**Figure 3.** Rule surface plots for dual-input fuzzy logic controllers 1 (**A**), 2 (**B**), 3 (**C**), 4 (**D**), and 5 (**E**).

### 2.4. Controller Performance Metrics

A total of 12 individual metrics were used to evaluate the performance of each fluid resuscitation controller during the simulated hemorrhage scenarios. Additionally, with the goal of making all these measurements more useful for reaching conclusions about the controllers, 3 aggregate combinations of the individual metrics were also calculated. All of these measurements have been described previously [5,11,27,28].

A number of the individual metrics used are derived from measurements of performance error (PE)—that is, the difference between the measured pressure at a given time and the target pressure, as a percentage of the target. In summary, these metrics were:

- Median performance error (MDPE): median value of all the PEs;
- Median absolute performance error (MDAPE): median of the absolute values of all the PEs;
- MDAPE at steady state ($MDAPE_{SS}$): MDAPE after system has reached steady state;
- Target overshoot: maximum positive PE value, relative to the target pressure;
- Effectiveness: percent of time that the pressure remained within 5 mmHg of the target value
- Wobble: median of the absolute values of the differences between each PE and MDPE;
- End-state divergence: expressed as a percentage, this is the slope of the linear regression of PE vs. time during the final 10% of the test scenario, multiplied by the total duration of the scenario;
- Percent rise time: amount of time required for the measured MAP to reach 90% of the target, relative to the total duration of the scenario;
- Volume efficiency: ratio of total volume of fluid infused over the output volume;
- Areas above and below target: expressed as a percentage, these are the total areas delimited by the target pressure line and the measured MAP-vs-time curve, both above and below said line, respectively, relative to the target pressure and further normalized by scenario time duration;
- Mean infusion rate: mean rate of infusion as a percentage of the maximum infusion rate allowed by the controller (500 mL/min);
- Infusion rate variability: the averaged standard deviations of the infusion rates as a percentage of the mean infusion rate.

The aggregate metrics derived from the aforementioned individual ones were used to aide in evaluating the controllers' overall performances in three areas, as follows:

- Intensity: the controller's ability to effectively treat hypotension; it is the product of Percent rise time and Area below target, divided by the Effectiveness.
- Stability: the controller's propensity for stable performance and reduced overshooting; it is the product of Wobble, the absolute value of End-state divergence, the squared value of $MDAPE_{SS}$, and the sum of Area above target and Target overshoot.
- Resource efficiency: the controller's capacity for reduced fluid consumption and hardware wearing; it is the product of Mean infusion rate, Infusion rate variability and Volume efficiency.

It should be noted that whenever any of the measurements listed above are evaluated, except for "Effectiveness", lower values are generally considered better.

### 2.5. Statistical Analysis

For each controller, three subject variability experiments were conducted for all the test scenarios. Each metric was made unitless as described in Section 2.4. Metrics were averaged across all test scenarios for each subject variability and normalized to the median value for each metric to make the weights for each metric similar. Aggregate metrics for Intensity, Stability, Resource Efficiency, and an average of each were calculated. Results throughout are reported as mean ± standard deviation. For evaluating statistical significance between aggregate scores, one-way analysis of variance (ANOVA) was used, post hoc Tukey's test, for each metric to evaluate differences between the five controllers. Significance was defined as $p < 0.05$.

### 3. Results

#### 3.1. Scenario 1: Low Initial MAP with Momentary Severe Hemorrhage Results

The first scenario tested began with a low MAP of 45 mmHg with no active hemorrhage. An intense hemorrhage was then introduced after 30 min, simulating a complication such as an extremity tourniquet failure, and lasted for 2 min. This was followed by an additional 30-min period with no active hemorrhage. This scenario evaluated the controllers' ability to resuscitate a patient without complications and test how quickly the controllers responded to an acute but brief hemorrhage. Plots of the MAP vs. Time and Flow Rate

vs. Time for a single run of each controller are presented in Figure 4A with positive flow rate values representing the infusion rate outputs of the controllers and negative flow rate values being a representative plot of the outflow (a combination of basal urine rate and hemorrhage). Percent Area Above Target and Absolute End-State Divergence are shown in Figure 4B,C, respectively, and results for all performance metrics can be found in Table A1. While all controllers achieved less than 5% Area Above Target relative to the target pressure and total scenario time, DFL 4 demonstrated the best overshoot performance in this scenario with a near 0% result. DFL 1 and 2 technically performed the worst, both overshooting around 3%. None of the controllers exceeded the overshoot limit of 5% of the target to cause a re-bleed event. DFL 1 and 2 had the lowest End-State Divergence relative to total scenario time with values of 0.10% and 0.15%, respectively. These both were significantly lower than the other three controllers which all were above 0.5% with DFL 5 having the highest value of 1.04%.

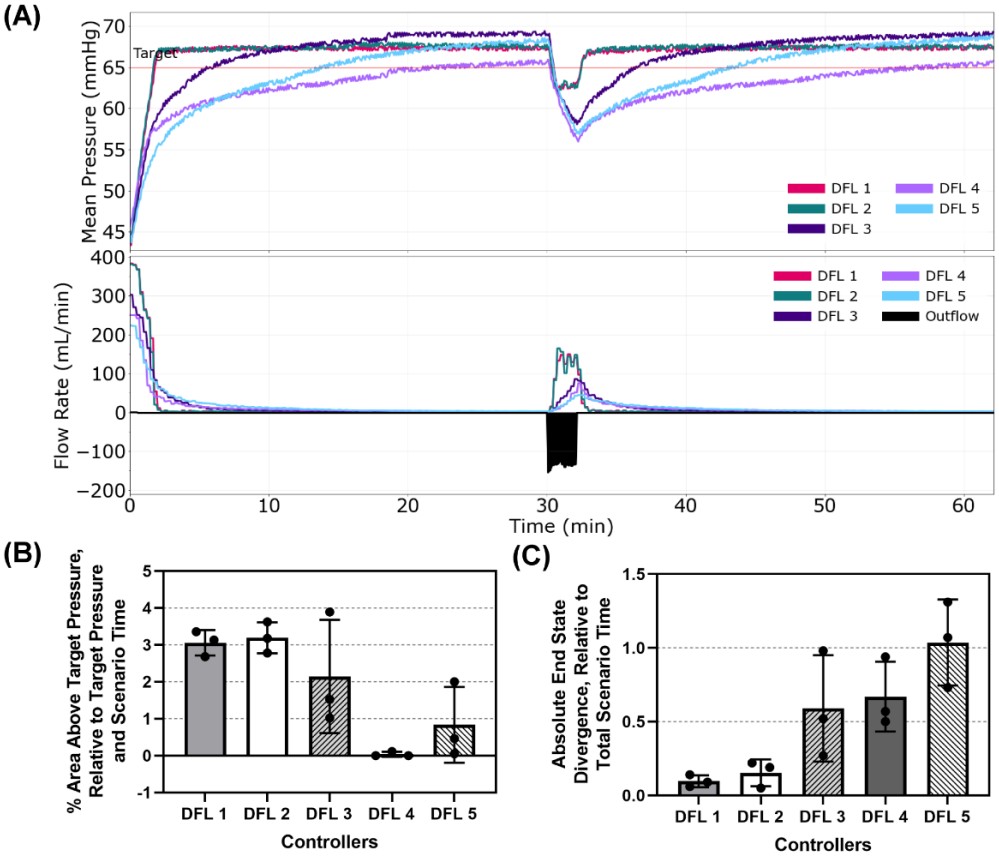

**Figure 4.** Dual-input fuzzy logic results for Scenario 1. Scenario 1 began with a MAP of 45 mmHg with no active hemorrhage for 30 min, followed by a fast hemorrhage for 2 min and then no hemorrhage for the remaining 30 min. (**A**) Five controller designs' MAP and infusion rate vs. time are shown for one replicate run. A single representative outflow vs. time result is shown. (**B**) Area above target pressure and (**C**) Absolute end state divergence performance metrics for each controller design are shown as mean values from three subject variability runs, with error bars denoting standard deviation.

## 3.2. Scenario 2: Target Initial MAP with Coagulating Hemorrhage Results

Scenario 2 presented the patient with a MAP starting at the targeted 65 mmHg but with an active hemorrhage that gradually reduced over time simulating an internal re-bleed accompanied by coagulation. This tested the controllers' responsiveness to perturbations to the system after reaching the set point. Plots of the MAP vs. Time and Flow Rate vs. Time for a single run of each controller are presented in Figure 5A. Percent Area Below Target and Percent Infusion Rate Variability are shown in Figure 5B,C, respectively, and results for all performance metrics can be found in Table A2. DFL 1 and 2 were the most responsive

to a drop in MAP while near the target pressure with % areas below target of 0.25% and 0.27%, respectively. DFL 3–5 all had significantly higher areas below the target with DFL 4 having the highest value (8.5%). It should be noted that DFL 1 and 2, as well as 5, ended up overshooting the target, though not enough to trigger a re-bleed penalty (Table A2). DFL 5 had the lowest % Infusion Rate Variability (7.24%) while DFL 1 had the highest (27.9%). This can be visually observed when looking at the varying magnitudes of the peaks in the Flow Rate vs. Time plot for DFL 1 and DFL 2 which had the second highest % Infusion Rate Variability (21.9%).

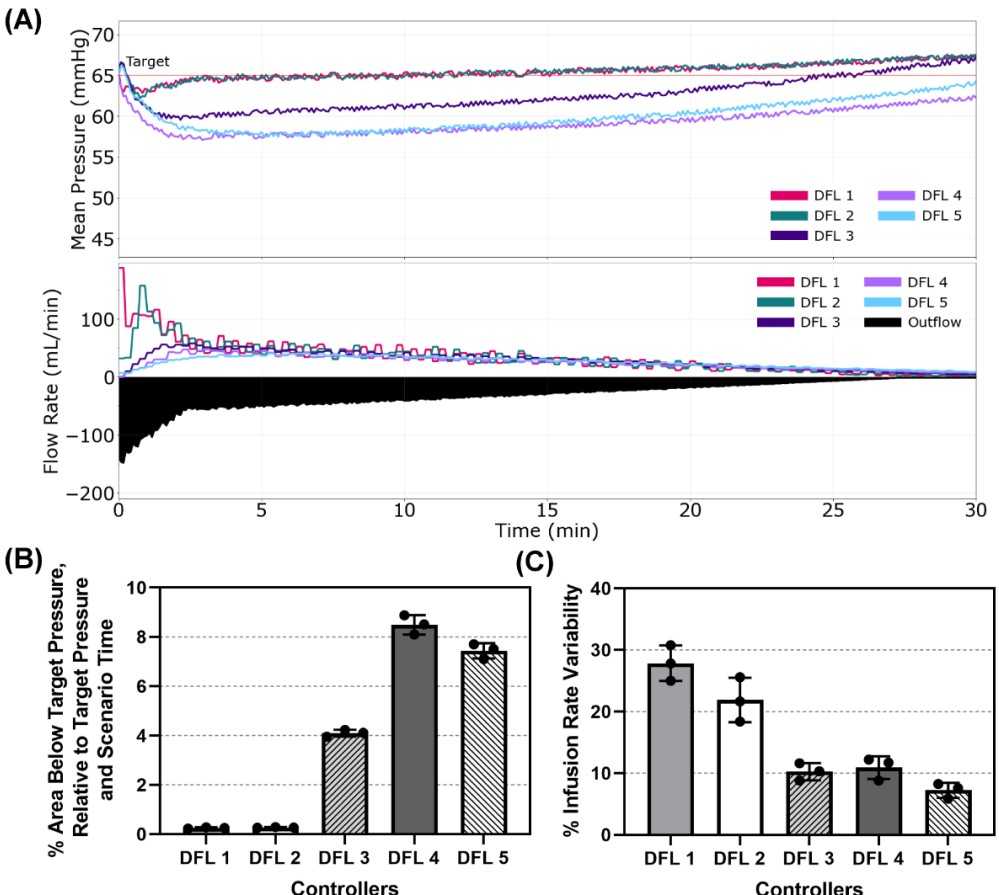

**Figure 5.** Dual-input fuzzy logic results for Scenario 2. Scenario 2 began with a MAP of 65 mmHg with an active hemorrhage that slows with time to mimic coagulation for 30 min, with a resuscitation target of 65 mmHg. (**A**) Five controller designs' MAP and infusion rate vs. time are shown for one replicate run. A single representative outflow vs. time result is shown. (**B**) Area below target pressure and (**C**) infusion rate variability performance metrics for each controller design are shown as mean values from three subject variability runs, with error bars denoting standard deviation.

### 3.3. Scenario 3: Low Initial MAP with Coagulating Hemorrhage Results

Scenario 3 began with a low MAP of 45 mmHg like Scenario 1 but included an ongoing hemorrhage with accompanying coagulation effects like Scenario 2. This scenario evaluated how effectively the controllers resuscitated a patient against complications like an internal, non-compressible hemorrhage. Plots of the MAP vs. Time and Flow Rate vs. Time for a single run of each controller are presented in Figure 6A. Percent Rise Time and % Effectiveness are shown in Figure 6B,C, respectively, and results for all performance metrics can be found in Table A3. DFL 4 and 5 had extremely high % Rise times compared to the other three controllers (44.7% and 41.3%, respectively) with DFL 1 and 2 performing almost identically with the lowest rise times (6.30% and 6.20%, respectively). This inversely correlates with the % Effectiveness with all 5 controllers maintaining the same relative

rankings with respect to each other (DFL 1 = 93.1%, DFL 2 = 92.9%, DFL 3 = 81.4%, DFL 5 = 24.5%, and DFL 4 = 34.0%).

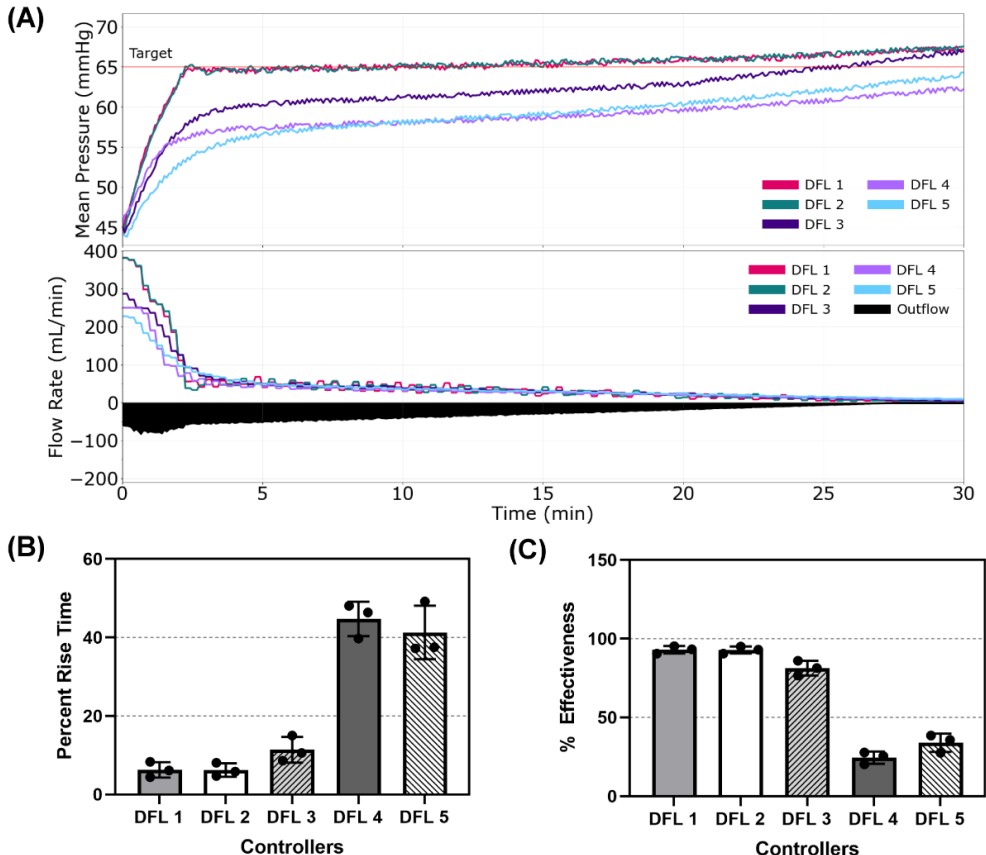

**Figure 6.** Dual-input fuzzy logic results for Scenario 3. Scenario 3 began with a MAP of 45 mmHg with an active hemorrhage slowing with time to mimic coagulation for 30 min, with a resuscitation target of 65 mmHg. (**A**) Five controller designs' MAP and infusion rate vs. time are shown for one replicate run. A single representative outflow vs. time result is shown. (**B**) Percent rise time and (**C**) effectiveness performance metrics for each controller design are shown as mean values from three subject variability runs, with error bars denoting standard deviation.

*3.4. Scenario 4: Low Initial MAP with Coagulopathic Hemorrhage*

Lastly, Scenario 4 provided the most complications of the scenarios tested. The patient began with a low MAP of 45 mmHg and presented with an ongoing non-compressible hemorrhage. This hemorrhage gradually slowed over time as the result of coagulation, but after 5 min, simulated coagulopathy was introduced gradually accelerating the hemorrhage until reaching a maximum rate of ~125 mL/min. This scenario was designed to tease out weaknesses of the controllers resulting in equilibrating infusion and outflow at a steady state that significantly deviates from the target. Plots of the MAP vs. Time and Flow Rate vs. Time for a single run of each controller are presented in Figure 7A. Percent Area Below Target and % MDAPE at Steady-State are shown in Figure 7B,C, respectively, and results for all performance metrics can be found in Table A4. These two metrics correlate closely in this scenario with the controller ranking and metric values nearly equal between the two. DFL 2 performed the best (3.56% area below target, 2.69% MDAPE at Steady State) and DFL 1 was nearly identical (3.56% area below target, 2.70% MDAPE at Steady State). DFL 3 (10.9% area below target, 10.8% MDAPE at Steady State) and DFL 4 (14.6% area below target, 14.4% MDAPE at Steady State) were next, and DFL 5 performed the worst in these two metrics (16.6% area below target, 16.8% MDAPE at Steady State).

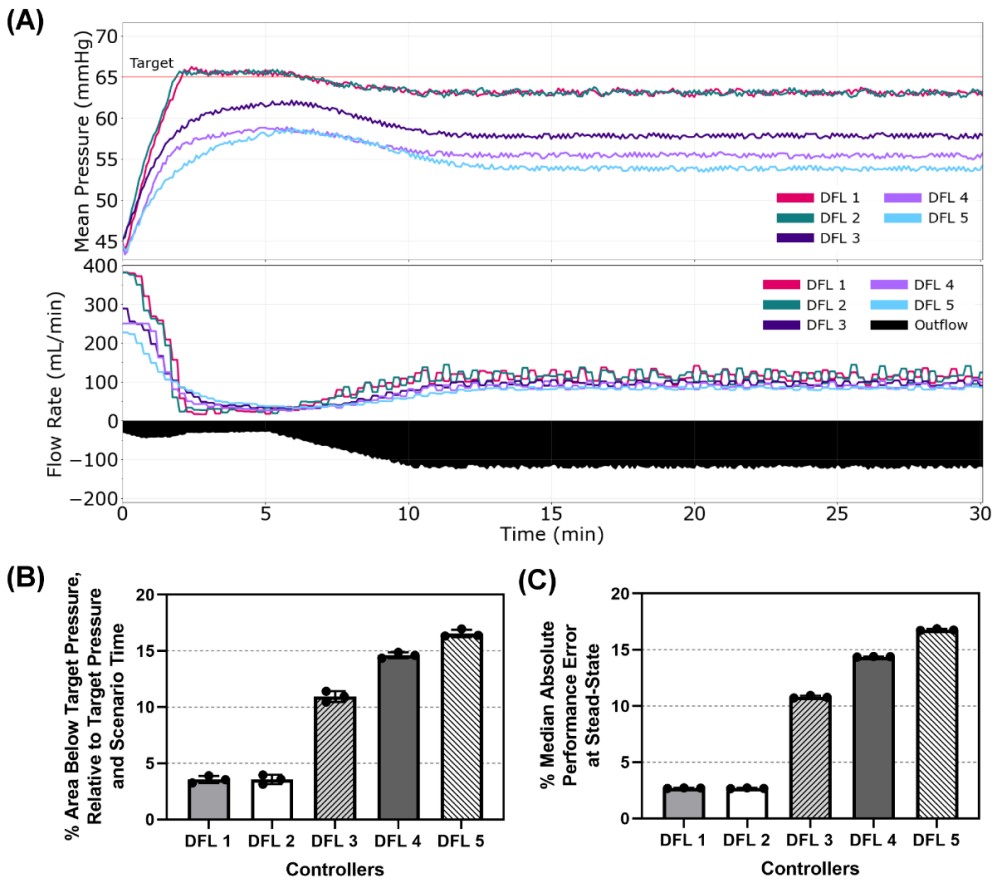

**Figure 7.** Dual-input fuzzy logic results for Scenario 4. Scenario 4 began with a MAP of 45 mmHg with an active hemorrhage slowing for the first 5 min and then accelerating to a maximum hemorrhage rate to mimic coagulopathy, with a resuscitation goal of 65 mmHg MAP. (**A**) Five controller designs MAP and infusion rate vs. time are shown for one replicate run. A single representative outflow vs. time result is shown. (**B**) The area below target pressure and (**C**) median absolute performance error at steady state performance metrics for each controller design are shown as mean values from three subject variability runs, with error bars denoting standard deviation.

### 3.5. Controller Performance in Aggregate Performance Metrics

We compiled the average score across all four scenarios of each aggregate performance metric for the 5 DFL controllers and then took the average of the three aggregate performance metrics (Figure 8). DFL 1 and 2 had the lowest two scores for the Intensity (both at 0.121) and Stability (0.314 and 0.299, respectively) metrics while holding the highest two scores in Resource Efficiency (2.02 and 1.88, respectively). DFL 4 had the highest score in the Intensity aggregate metric (10.18) with the second lowest score in Resource Efficiency (0.836). DFL 5 had the second highest score for Intensity (8.84) and held the lowest score for Resource Efficiency (0.604), though it had the highest score in the Stability aggregate (5.08). DFL 2 had the lowest average aggregate score (0.765) followed closely by DFL 1 (0.817) while DFL 4 had the second highest (3.88) and DFL 5 had the highest average aggregate (4.84). DFL 3 did not have the highest nor lowest of any aggregate score and had the median average aggregate score of the 5 controllers (1.49). A summary of one-way ANOVA statistical analyses comparisons for each aggregate metrics are shown in Table A5.

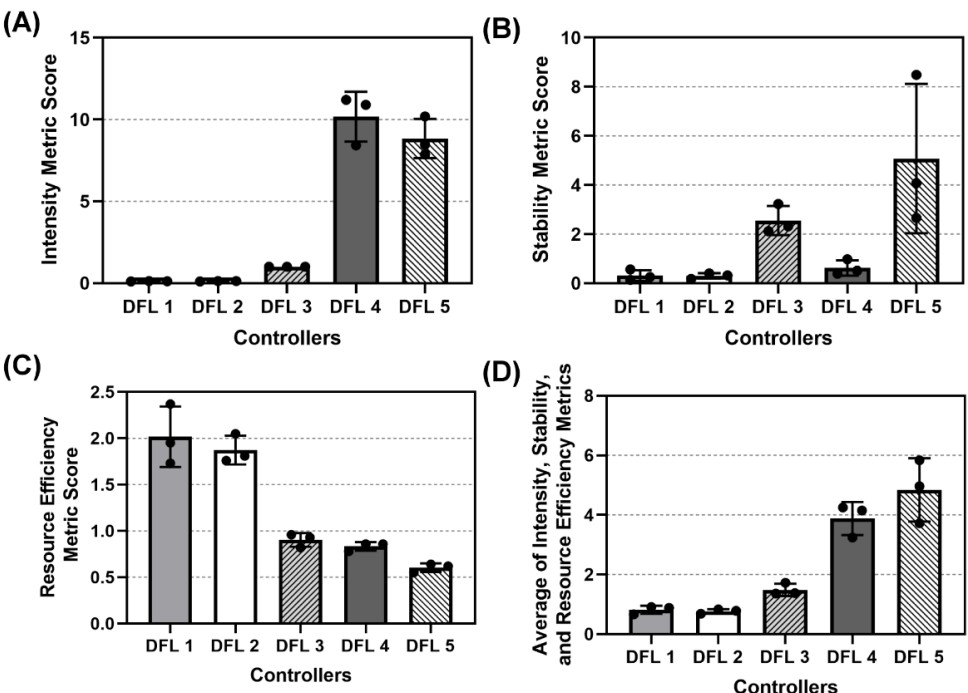

**Figure 8.** Aggregate metric results for dual-input fuzzy logic controller configurations. Aggregate (**A**) Intensity, (**B**) Stability, (**C**) Resource Efficiency, and (**D**) Average across the three metric results are shown as mean values from three subject variability runs, with error bars denoting standard deviation.

## 4. Discussion

Hemorrhagic shock resuscitation remains a challenging aspect of emergency medicine and trauma care. This is especially true in the context of resource and care provider-limited environments or mass casualty incidents where the attention needed to properly monitor their status and adjust therapy accordingly can quickly exceed current capabilities. The implementation of automated systems to provide DCR in these environments can lighten this load on care providers and potentially improve patient outcomes. Previously, multiple physiological closed-loop controllers were evaluated, and DFL controllers were shown to have the best performance in an assortment of performance and aggregated metrics [11].

This work compared an expanded set of five DFL controllers to identify which one performed favorably using the HATRC system. Three aggregate metrics were used to evaluate the controllers along the criteria of Intensity, Stability, and Resource Efficiency. Four hemorrhage scenarios were used in the comparison to assess how the controllers performed against an array of patient conditions including low MAP, with and without ongoing hemorrhage, the introduction of sudden, acute hemorrhage, and the introduction of ongoing hemorrhage. We also tested a scenario particularly designed to challenge the controllers' ability to overcome steady state error.

DFL 1 and 2 had the best performance in Intensity and Stability, indicating they were best suited to quickly reach the set point with the lowest degree of long-term oscillations in MAP. Although they tended to overshoot the target, especially when there was no ongoing hemorrhage, they did not exceed the overshoot threshold allowed, and the amount of overshoot was minimal with respect to the other metrics considered when calculating Stability. They performed worst in Resource Efficiency indicating they placed a high demand on equipment and consumed the most amount of fluid. This is understandable when observing the larger peak-to-trough magnitudes in infusion rate and the fact that a contributor to hemorrhage rate in HATRC was the MAP—i.e., sustaining a higher MAP for a larger proportion of the scenario time resulted in a larger accumulative hemorrhage, requiring more infused volume to compensate. DFL 4 and 5 showed the worst performance in Intensity but had the best scores in Resource Efficiency. This illustrates the compromise

taking place between fast, aggressive resuscitation aimed at rapid restoration of oxygen delivery to the tissues, and a more gradual approach. It also emphasizes the gap in current clinical knowledge regarding which method is optimal for patients' outcomes. There may be conditions that would be better treated by one versus the other, but there has not been a robust enough study to draw conclusions based on established patient outcomes. All that said, in this study DFL 2 had the best average score in our tests, seemingly offering the most comprehensive balance of the metrics evaluated. This controller will be further investigated for potential optimizations and considered for testing in other models.

The current study does possess certain limitations that should be considered. The HATRC platform was designed based on empirical data and does not contain the degree of complexity and unpredictability of an in vivo model. There are alternative models, such as in silico simulations that may be useful for further evaluating the capabilities of automated controllers [29,30]. The intent here was to provide real-world performance data when physical hardware was used, and we believe the results show promise within the limitations of the empirical data-guided platform. The controllers investigated used MAP as the sole input, but real-time values provided by invasive measurements may not always be available. The reduced feedback frequency of current non-invasive arterial blood pressure measurement techniques may greatly hinder the performance of these controllers. Improvement opportunities exist in using other physiological variables for inputs as an alternative, such as cardiac output [31], a photoplethysmography waveform [32], or tissue oxygen saturation. Although the membership functions and their corresponding parameters were selected based on feedback from subject matter experts in anesthesiology, surgery, and military medicine, the lack of universal agreement within the medical community on the best resuscitation profile and the unpredictable nature of the physiologic response makes this tuning difficult and requires further refinement. This could be offered by more complex models that cover a wider range of physiologic states, such as septic shock, which would also make it possible to expand into type-2 fuzzy logic systems and iteratively optimize using simulation techniques [33–35]. As previously mentioned, the aggregate performance metrics used do not account for the full scope of physiological responses present when systemic trauma is experienced and may require weighting of some metrics over others. There are also unknowns regarding multiple simultaneous injuries, incapacitation of certain physiological systems, and the impact of chemical therapies that may be present. These interactions and their potential effect on arterial blood pressure, such as distributive or cardiogenic shock, were outside the scope of the current study but will be addressed in future in vivo ones.

## 5. Conclusions

Hemorrhagic shock goal-directed resuscitation can be facilitated in both emergency and military medicine by automating the constant fluid rate changes required to adequately stabilize a patient. Dual input fuzzy logic controllers have a wide range of tunability for managing various hemorrhagic shock resuscitation scenarios. As advancements in hemorrhagic shock resuscitation standard of care develop, DFL controllers have demonstrated the flexibility to be adapted to meet physiological demands that can promote the most desired patient outcomes. Through aggregate performance metric scores, a single DFL controller was identified as performing best which will be further evaluated in large animal hemorrhagic shock studies. This will bring closed loop control for acute hemorrhage resuscitation closer to reality to help improve the patient's recovery and stabilization while lessening the cognitive burden for the medical provider.

**Author Contributions:** Conceptualization, D.B., S.J.V., E.J.S. and G.A.; methodology, D.B. and E.J.S.; data collection, D.B. and S.J.V.; formal analysis D.B., E.J.S. and S.J.V.; writing (original draft preparation), D.B., S.J.V., E.J.S. and G.A.; writing (review and editing), D.B., S.J.V., E.J.S. and G.A.; supervision and project administration, E.J.S. All authors have read and agreed to the published version of the manuscript.

**Funding:** This work was funded by the U.S. Army Medical Research and Development Command (Proposal Number IS220008).

**Institutional Review Board Statement:** Not applicable.

**Informed Consent Statement:** Not applicable.

**Data Availability Statement:** The datasets generated during and/or analyzed during the current study are available from the corresponding author upon reasonable request.

**Conflicts of Interest:** The authors declare no conflict of interest.

**DoD Disclaimer:** The views expressed in this article are those of the authors and do not reflect the official policy or position of the U.S. Army Medical Department, Department of the Army, DoD, or the U.S. Government.

## Appendix A

**Table A1.** Summary of performance metrics for Scenario 1. Performance metrics for each of five DFL controllers is shown as mean values for three subject variability runs.

|  | DFL 1 | DFL 2 | DFL 3 | DFL 4 | DFL 5 |
|---|---|---|---|---|---|
| MDPE (%) | 3.25% | 3.41% | 2.18% | −4.04% | −1.15% |
| MDAPE (%) | 3.32% | 3.46% | 3.16% | 4.04% | 3.12% |
| MDAPE_SS (%) | 3.29% | 3.43% | 3.03% | 3.17% | 2.42% |
| Target Overshoot (%) | 4.25% | 4.59% | 5.00% | 0.46% | 3.79% |
| Effectiveness (%) | 97.14% | 97.23% | 94.01% | 83.29% | 86.06% |
| Wobble (%) | 0.43% | 0.42% | 0.94% | 1.20% | 1.18% |
| End-State Divergence (%) | 0.10% | 0.15% | 0.59% | 0.67% | 1.04% |
| Percent Rise Time (%) | 2.64% | 2.64% | 4.12% | 6.02% | 8.47% |
| Volume Efficiency | 310.93% | 311.83% | 321.43% | 297.47% | 312.70% |
| Area Above Target Pressure (%) | 3.06% | 3.19% | 2.14% | 0.04% | 0.84% |
| Area Below Target Pressure (%) | 0.73% | 0.72% | 1.73% | 5.21% | 3.80% |
| Mean Infusion (%) | 3.72% | 3.75% | 3.77% | 3.26% | 3.58% |
| Variable Infusion (%) | 40.34% | 41.03% | 22.27% | 24.58% | 16.72% |

**Table A2.** Summary of performance metrics for Scenario 2. Performance metrics for each of five DFL controllers is shown as mean values for three subject variability runs.

|  | DFL 1 | DFL 2 | DFL 3 | DFL 4 | DFL 5 |
|---|---|---|---|---|---|
| MDPE (%) | 0.55% | 0.56% | −4.61% | −9.03% | −8.07% |
| MDAPE (%) | 0.94% | 0.93% | 4.61% | 9.03% | 8.07% |
| MDAPE_SS (%) | 0.94% | 0.93% | 3.60% | 8.27% | 6.78% |
| Target Overshoot (%) | 3.54% | 3.52% | 1.70% | 0.51% | 1.30% |
| Effectiveness (%) | 100.28% | 100.28% | 98.61% | 32.04% | 46.67% |
| Wobble (%) | 0.90% | 0.92% | 1.39% | 1.14% | 1.20% |
| End-State Divergence (%) | 0.54% | 0.51% | 1.06% | 1.15% | 1.72% |
| Percent Rise Time (%) | NA | NA | NA | NA | NA |
| Volume Efficiency | 105.00% | 102.70% | 97.67% | 82.87% | 87.50% |
| Area Above Target Pressure (%) | 0.88% | 0.92% | 0.09% | 0.00% | 0.01% |
| Area Below Target Pressure (%) | 0.25% | 0.27% | 4.10% | 8.50% | 7.44% |
| Mean Infusion (%) | 6.69% | 6.55% | 5.72% | 4.52% | 4.86% |
| Variable Infusion (%) | 27.87% | 21.88% | 10.25% | 10.91% | 7.24% |

**Table A3.** Summary of performance metrics for Scenario 3. Performance metrics for each of five DFL controllers is shown as mean values for three subject variability runs.

|  | DFL 1 | DFL 2 | DFL 3 | DFL 4 | DFL 5 |
|---|---|---|---|---|---|
| MDPE (%) | 0.45% | 0.55% | −4.88% | −9.96% | −9.50% |
| MDAPE (%) | 1.06% | 1.07% | 4.88% | 9.96% | 9.50% |
| MDAPE_SS (%) | 0.92% | 0.92% | 1.89% | 7.44% | 5.36% |
| Target Overshoot (%) | 3.49% | 3.54% | 1.48% | 0.00% | 0.00% |
| Effectiveness (%) | 93.06% | 92.87% | 81.39% | 24.54% | 33.98% |
| Wobble (%) | 0.92% | 0.89% | 1.25% | 1.11% | 1.16% |
| End-State Divergence (%) | 0.49% | 0.54% | 1.16% | 1.41% | 1.36% |
| Percent Rise Time (%) | 6.30% | 6.20% | 11.39% | 44.72% | 41.30% |
| Volume Efficiency | 186.47% | 188.07% | 191.40% | 179.70% | 192.83% |
| Area Above Target Pressure (%) | 0.88% | 0.92% | 0.09% | 0.00% | 0.00% |
| Area Below Target Pressure (%) | 1.67% | 1.65% | 5.91% | 10.39% | 10.47% |
| Mean Infusion (%) | 10.83% | 10.96% | 9.88% | 8.63% | 8.96% |
| Variable Infusion (%) | 23.39% | 21.76% | 13.36% | 15.42% | 11.34% |

**Table A4.** Summary of performance metrics for Scenario 4. Performance metrics for each of five DFL controllers is shown as mean values for three subject variability runs.

|  | DFL 1 | DFL 2 | DFL 3 | DFL 4 | DFL 5 |
|---|---|---|---|---|---|
| MDPE (%) | −2.76% | −2.75% | −10.86% | −14.43% | −16.85% |
| MDAPE (%) | 2.76% | 2.75% | 10.86% | 14.43% | 16.85% |
| MDAPE_SS (%) | 2.70% | 2.69% | 10.81% | 14.38% | 16.79% |
| Target Overshoot (%) | 1.18% | 0.94% | 0.00% | 0.00% | 0.00% |
| Effectiveness (%) | 93.54% | 93.63% | 15.51% | 0.00% | 0.00% |
| Wobble (%) | 0.47% | 0.48% | 0.58% | 0.55% | 0.68% |
| End-State Divergence (%) | 0.18% | 0.09% | 0.16% | 0.11% | 0.20% |
| Percent Rise Time (%) | 5.83% | 5.93% | 9.54% | 15.42% | 19.44% |
| Volume Efficiency | 120.67% | 121.13% | 117.40% | 117.43% | 113.60% |
| Area Above Target Pressure (%) | 0.06% | 0.05% | 0.00% | 0.00% | 0.00% |
| Area Below Target Pressure (%) | 3.56% | 3.56% | 10.94% | 14.61% | 16.56% |
| Mean Infusion (%) | 22.87% | 22.92% | 18.92% | 17.35% | 15.91% |
| Variable Infusion (%) | 14.32% | 14.99% | 9.32% | 11.51% | 8.05% |

**Table A5.** Summary of statistical analysis for aggregate metrics averaged across all tested scenarios. $p < 0.05$ indicated statistical significance. Values are italicized when this threshold was reached for the particular comparison pairing.

| | **Statical Analysis for Intensity Aggregate Scores** | | | | |
|---|---|---|---|---|---|
| | **DFL 1** | **DFL 2** | **DFL 3** | **DFL 4** | **DFL 5** |
| DFL 1 | | | | | |
| DFL 2 | > 0.99 | | | | |
| DFL 3 | 0.7297 | 0.7323 | | | |
| DFL 4 | *<0.0001* | *<0.0001* | *<0.0001* | | |
| DFL 5 | *<0.0001* | *<0.0001* | *<0.0001* | 0.3829 | |
| | **Statical Analysis for Stability Aggregate Scores** | | | | |
| | **DFL 1** | **DFL 2** | **DFL 3** | **DFL 4** | **DFL 5** |
| DFL 1 | | | | | |
| DFL 2 | >0.9999 | | | | |
| DFL 3 | 0.3469 | 0.3405 | | | |
| DFL 4 | 0.9986 | 0.9983 | 0.4788 | | |
| DFL 5 | *0.0128* | *0.0125* | 0.2509 | *0.0194* | |
| | **Statical Analysis for Resource Efficiency Aggregate Scores** | | | | |
| | **DFL 1** | **DFL 2** | **DFL 3** | **DFL 4** | **DFL 5** |
| DFL 1 | | | | | |
| DFL 2 | 0.8266 | | | | |
| DFL 3 | *<0.0001* | *0.0002* | | | |
| DFL 4 | *<0.0001* | *0.0001* | 0.9841 | | |
| DFL 5 | *<0.0001* | *<0.0001* | 0.2548 | 0.4824 | |
| | **Statical Analysis for Average Aggregate Scores** | | | | |
| | **DFL 1** | **DFL 2** | **DFL 3** | **DFL 4** | **DFL 5** |
| DFL 1 | | | | | |
| DFL 2 | 0.6678 | | | | |
| DFL 3 | 0.2177 | 0.1343 | | | |
| DFL 4 | *0.043* | *0.0384* | *0.0426* | | |
| DFL 5 | 0.0726 | 0.0705 | 0.0941 | 0.2818 | |

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
