# Peer review of "Dual Input Fuzzy Logic Controllers for Closed Loop Hemorrhagic Shock Resuscitation"

_processes, doi:10.3390/pr10112301_

Round 1

Reviewer 1 Report

In the manuscript, four simulated hemorrhage scenarios are studied to evaluate the performance of five fuzzy logic controllers. Please consult the literature on fuzzy logic control, there are no the references in the manuscript. Unfortunately, in a Section "... Methods", the manuscript looks like a work of a student who uses a MATLAB Fuzzy Logic Designer toolbox but does not understand the methods fully. But my main unfavourable comment is the following. There are no grounds for a choice of membership functions, it seems arbitrary.

However, based on advantages of fuzzy logic control, I am willing to give the authors an opportunity to improve the revision.

Author Response

We thank the reviewer for carefully reviewing our manuscript submission. We have responded to each comment/question point by point below and have tracked changes in the resubmitted manuscript draft. We hope the manuscript is now suitable for publication in Processes:

In the manuscript, four simulated hemorrhage scenarios are studied to evaluate the performance of five fuzzy logic controllers. Please consult the literature on fuzzy logic control, there are no the references in the manuscript. Unfortunately, in a Section "... Methods", the manuscript looks like a work of a student who uses a MATLAB Fuzzy Logic Designer toolbox but does not understand the methods fully. But my main unfavourable comment is the following. There are no grounds for a choice of membership functions, it seems arbitrary.

However, based on advantages of fuzzy logic control, I am willing to give the authors an opportunity to improve the revision.

The authors would like to thank the reviewer for the opportunity to improve this manuscript, making it more useful for this interesting field of research. We have included a paragraph at the beginning of Section 2.3 outlining some fundamentals of fuzzy logic control and directing readers to the literature illustrating some of its many uses.

Some explanation of the rationale behind the membership function selection has been added in lines 138-143 with additional references. As membership function selection is highly subjective and infinitely variable depending on the immediate application, as well as the system under consideration being extremely complex without a fully defined mathematical representation, the authors recognize the limitations currently faced and have elaborated on these further in the Discussion (line 395-402). Although the current configuration is not claimed to be the best, we believe the work clearly demonstrated the performance differences in the controllers evaluated and positions fuzzy logic control as a strong candidate for further development as clinical knowledge advances.

Reviewer 2 Report

Application identified by the authors is relevant and important. However author need to revise the paper

1. To provide more clarity on the need for using fuzzy system as compared to other techniques

2. Detailed discussion to be presented indicating, how the membership function have been considered. 

3. Clarity on the relevance on application.

Author Response

We thank the reviewer for carefully reviewing our manuscript submission. We have responded to each comment/question point by point below and have tracked changes in the resubmitted manuscript draft. We hope the manuscript is now suitable for publication in Processes:

Application identified by the authors is relevant and important. However author need to revise the paper

  1. To provide more clarity on the need for using fuzzy system as compared to other techniques

We chose fuzzy logic controllers based on a previous comparative study conducted on decision table, PID, and two different fuzzy inference systems (line 117-122). The performance results favored the dual-input fuzzy logic controllers which led to the current study.

  1. Detailed discussion to be presented indicating, how the membership function have been considered. 

The authors have elaborated on the rationale behind the selection of the membership functions in lines 138-143. Due to the highly subjective nature of this selection as well as the system under consideration lacking a fully defined mathematical representation, we have further elaborated on these limitations in the Discussion (line 395-402).

  1. Clarity on the relevance on application.

The relevance of the application is automating hemorrhagic shock resuscitation. We have added a statement about how this requires constant fluid rate adjustments, to further justify the relevance of closed loop systems for this application. We have previously developed a number of closed loop controllers and identified the dual input fuzzy logic controller was optimal. As a result in this manuscript we are evaluating tuning of this single closed loop controller type.                                  

Reviewer 3 Report

The article is very interesting and has a practical importance to the field . It is well presented and established. There are a few papers that need to be cited when presenting FS, for instance:

Total ordering defined on the set of all intuitionistic fuzzy numbers

V Nayagam, S Jeevaraj, G Sivaraman Journal of Intelligent & Fuzzy Systems 30 (4), 2015-2028

Author Response

We thank the reviewer for carefully reviewing our manuscript submission. We have responded to each comment/question point by point below and have tracked changes in the resubmitted manuscript draft. We hope the manuscript is now suitable for publication in Processes:

The article is very interesting and has a practical importance to the field. It is well presented and established. There are a few papers that need to be cited when presenting FS, for instance:

Total ordering defined on the set of all intuitionistic fuzzy numbers  - V Nayagam, S Jeevaraj, G Sivaraman Journal of Intelligent & Fuzzy Systems 30 (4), 2015-2028

The authors thank the reviewer for this useful reference and have included it along with others in the Discussion section.

Reviewer 4 Report

In the manuscript, there are compared five dual-input fuzzy logic controllers with different membership function to identify which one performed favorably using the HATRC (Hardware-in-loop Automated Testbed for Resuscitation Controllers) platform. Four hemorrhage scenarios and three aggregate metrics were used to evaluate the controllers along the criteria of Intensity, Stability, and Resource Efficiency. Based on the obtained results, it seems that dual-input fuzzy logic controllers are well-suited for hemorrhagic shock resuscitation and can be tuned to meet the response rates set by clinical practice guidelines for this application.

Comments and Questions:

1. It would be fine to add a block diagram of the control scheme.

2. In the Subsection 2.1 there is written, that MATLAB was used to run the hemorrhage scenario, , determine inflow rates based on resuscitation controller algorithms, and control the corresponding pumps through an RS-232 serial connection. What hardware and baud rate was used?

3. In the Subsection 2.3 there is written, that all controllers used the same fuzzy sets and membership function types for (d/dt)PerformanceError. But the membership function in Fig. 1D seems to be a bit different.

4. Maybe it would be better to include Tables  S1-5 from Supplementary Information in the manuscript as Appendix, because they are important for readers for comparison of obtained results.

Author Response

We thank the reviewer for carefully reviewing our manuscript submission. We have responded to each comment/question point by point below and have tracked changes in the resubmitted manuscript draft. We hope the manuscript is now suitable for publication in Processes:

In the manuscript, there are compared five dual-input fuzzy logic controllers with different membership function to identify which one performed favorably using the HATRC (Hardware-in-loop Automated Testbed for Resuscitation Controllers) platform. Four hemorrhage scenarios and three aggregate metrics were used to evaluate the controllers along the criteria of Intensity, Stability, and Resource Efficiency. Based on the obtained results, it seems that dual-input fuzzy logic controllers are well-suited for hemorrhagic shock resuscitation and can be tuned to meet the response rates set by clinical practice guidelines for this application.

Comments and Questions:

  1. It would be fine to add a block diagram of the control scheme.

Great suggestion. We have added a block diagram for the dual input fuzzy logic controller as Figure 1 in the manuscript.

  1. In the Subsection 2.1 there is written, that MATLAB was used to run the hemorrhage scenario, , determine inflow rates based on resuscitation controller algorithms, and control the corresponding pumps through an RS-232 serial connection. What hardware and baud rate was used?

We have added a description of the connection hardware in the manuscript at line 79 in the methods. Baud rate was set at 4800, but this was just the setting indicated by the pump manufacturer for the pumps we were using. We indicated this as well in the methods.

  1. In the Subsection 2.3 there is written, that all controllers used the same fuzzy sets and membership function types for (d/dt)PerformanceError. But the membership function in Fig. 1D seems to be a bit different.

The authors would like to thank the reviewer for this detailed observation. Although the fuzzy sets and corresponding membership function types were the same for (d/dt)PerformanceError (linear z-shaped, triangular, and linear s-shaped), it is correct that there are slight differences in the plots shown in Figure 1. This is because the controllers had differences in the parameters used for their membership functions based on the desired performance which was explained in lines 153-156.

  1. Maybe it would be better to include Tables  S1-5 from Supplementary Information in the manuscript as Appendix, because they are important for readers for comparison of obtained results.

Great suggestion, we have moved the supplementary tables into Appendix A, and updated references throughout the text.

Round 2

Reviewer 1 Report

no

Reviewer 2 Report

Authors have satisfactorily revised the paper

Reviewer 4 Report

Dear Authors,

The manuscript can be accepted for publication.

 Reviewer